# A Software Products Line as Educational Tool to Learn Industrial Robots Programming with Arduino

**Andrés Felipe Solis Pino** [1,2,*] **, Pablo H. Ruiz** [1] **and Julio Ariel Hurtado Alegria** [2]

1  Facultad de Ingeniería, Corporación Universitaria Comfacauca—Unicomfacauca, Popayán 190003, Colombia; pruiz@unicomfacauca.edu.co

2  Facultad de Ingeniería Electronica y Telecomunicaciones, Universidad del Cauca, Popayán 190003, Colombia; ahurtado@unicauca.edu.co

*  Correspondence: asolis@unicomfacauca.edu.co

**Abstract:** Software reuse has potential for educational purposes since it uses decomposition and abstraction, two necessary skills to learn programming. Software reuse techniques require abstractions that are not obvious to students or even to professionals. Taking advantage of these techniques, students can learn computer programming in a productive and organized way. This paper proposes to use the Software Product Line (SPL) reuse technique as a strategy for learning to program industrial robots with the Arduino platform. First, the paper explains SPL construction and application with first-year university students. The SPL proposes abstractions close to the industrial robots domain with a simplified variability. The paper uses the case study method to show the feasibility of using the SPL approach in a learning environment. In this evaluation, students reused 38% to 43% of the total code needed to program the robot. It represents an improvement in the time it takes students to program industrial robotics solutions facilitating their learning. In addition, the paper unveils some limitations related to usability, specific knowledge, and some exploitable technologies.

**Keywords:** software product lines; educational robotic; industrial robots; Arduino

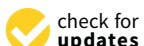

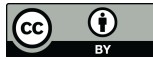

## 1. Introduction

Industrial Robotic Systems (IRS) are manipulative, functional, and programmable devices for working with objects according to defined trajectories to perform various tasks [1]. Nowadays, programming these devices is imperative because they play a more relevant role in the technology industry. Several domains, such as the manufacturing industry, the automotive industry, and the health sciences [2], are witnesses to the multiple advantages of the use of industrial robots. Moreover, areas such as the naval industry and search and rescue operations [3] are examples of the diversification of tasks that these devices can fulfill. The above ideas allow us to affirm that the IRS has increased importance in society, requiring universities to work toward new training programs and educational strategies in this area of knowledge. In particular, the construction of IRS in an educational environment needs the realization of prototypes, allowing students to understand the domain and its main theoretical and practical foundations [4]. These prototypes usually use microcontrollers (Arduino) to provide the functionality to electronic devices [5], which allows to emulate software development in the industry.

Traditional IRS software construction methodologies are key tools in engineering education; however, these are limited. For instance, they did not frequently unify hardware and software design as co-design approaches suggest [6]. It is widely accepted that novel approaches and reuse strategies should apply to different domains with enormous benefits [7]. Software Engineering (SE) and specifically software reuse have emerged as approaches to improving the development of IRS, but there are few cases where these techniques are replicable [8]. Model-Driven Engineering (MDE), Component-Based Software Engineering (CBSE), Service Oriented Architecture (SOA), and software product lines are the main reuse approaches applied in the area. The MDE, CBSE, and SOA are reuse approaches

that respectively segment the software of IRS into models, components, or services, intending to reduce the complexity in the applications, which facilitates the implementation of the software reuse and maintainability of domain assets [9,10]. However, engineers and programmers are resistant to the use of these approaches, which generate rework in software development, because the developers are unknown or because they do not trust the strategies to be proven and successful [11]. This low adoption may be because most IRs clients are companies that prefer reliability rather than other quality attributes, such as modifiability, as the key quality attribute [12]. Another reason for the adoption problem is the high variability in the hardware and software components, making the application of these techniques difficult [13], from domain abstraction, pattern detection, and idiom identification to characterization of granularity reuse level [8]. Moreover, general proposals have emerged covering many problems in the IRS, but they do not solve typical problems of the software device [14]. One of the reuse alternatives that could help to adopt software reuse is the SPL paradigm because it systematically defines a set of software products that share common and variable assets, which are configured and reused in a planned way to satisfy the needs of a specific domain [15].

Encouraging and applying reuse approaches to improve the software development process for microcontroller-based SRI helps benefit developers, particularly students, from planned reuse as well as understand and become familiar with these SE approaches from their academic background. In addition, it is important to note that there is a recent surge of interest from the research community in the contribution of robotics to the development of computational problem-solving thinking and skills, especially during the last decade. The development of new technologies and their adaptation to the school context has allowed the establishment of initiatives and projects to enhance new skills and competencies among the participants. These projects are viable with the use of robotics [16].

This paper aims to expose, through a case study, the potential of a proposed SPL-based software reuse approach on the IRS with Arduino, called IRArduino-SPL as a strategy for educational purposes using two powerful skills from the computational thinking area: abstraction and modularity.

The paper is organized as follows. The second section presents the major results of a literature review in the domain of IRS. The third section describes the materials and methods used in the research, including a study of the results found in SPL basic construction. The four and five sections allow us to discuss and conclude the proposed strategy and present possible future work in this direction.

## 2. Related Work

### 2.1. Software Reuse in Industrial Robots with Controllers

In previous work [10], the application of reuse approaches in IRS with controllers was analyzed. In addition, the MDE, CBSE, SOA, and SPL approaches were identified as the most used. The major challenges found were, among others, the variability of the hardware and the combination of reuse approaches to take advantage of each of them.

Brugali and Hochgeschwender [17] mention that SPLs have proven to be an efficient approach to face reuse in IRS, proposing a product line with a tool called Hyperflex, which uses the MDE to abstract these devices, thus supporting the entire software development cycle for robots with controllers. Finally, the authors conclude that the proposal allowed the structure and organization of the development of robots.

Gherardi et al. [18] propose to combine SPLs and cloud technologies to reuse robots with controllers in a tool called Rapyuta's which allows the variability to be modeled under three major elements: architecture variability model, variability of functionalities, and deployment variability model. As part of the validation of this proposal, they identified as necessary to model the constraints of the environment and the functionalities of the robot.

The SPLs have not only been implemented to build the software for IRS, but Abdelhady also proposed a different orientation [19], where he proposes an SPL with a framework for the development of trajectories and mappings. This work analyzes the difficulty in developing this type of system, which integrates hardware and software elements in real-

time. The results show an improvement in modularity, reduction of time, development effort, and deployment of these specific functionalities in the industrial robots with Arduino.

### 2.2. Arduino Implemented in the Robotics Domain

Multiple proposals have been developed between robotics and the Arduino platform, where these types of microcontrollers are used as low-cost alternatives to offer robotic solutions that work in different domains and provide various functionalities. This is the case of Concha Sánchez et al. in [20] where they present a methodology to recycle and update a robotic arm of 4 degrees of freedom in an educational context using elements such as artificial vision and decentralized control. To acquire the data from the environment, they use two Arduino Mega 2560 boards, and as the main controller, they use MATLAB software, getting as the main result a functional robotic arm of low-cost and open source.

Another related research is that of Marsono et al. [21] in which they propose to design and build a robotic arm to teach the GrblGru programming language (G-code) and Arduino as an application platform. The major result of this work was related to the creation of a functional prototype that helps to study the robotics programming language. The generalization of Arduino is such that it is even used in other branches of robotics, such as mobile, and in different domains such as agriculture [22], garbage collection [23], and gerontechnology [24], among others.

Finally, the above are examples of the multiple research carried out linking robotics and Arduino, so solutions focused on this domain that allow for improved software development on the platform would positively affect generalization and allow developers to focus on other hardware issues.

### 2.3. Software Engineering on the Arduino Platform

In the review by Geraldi et al. [25], they present an overview about SPLs application to devices with microcontrollers for the Internet of Things, where the implementations are made in different devices such as Raspberry Pi and Arduino. The results show the lack of the systematic and detailed specification of the SPLs that guarantee the quality of the derived products, as well as adaptation guidelines for their use.

Likewise, Bonfanti et al. [26] propose a code generator with MDE, from user specifications, using abstract state machines. This method guides the software development for real-time systems, based on the principle of refinement, thus capturing the semantics of the systems to the desired level. As part of the proposal, they developed the Asm2C++ tool using the MDE and model-to-text transformation (M2T) to generate code for microcontrollers. In addition, they also carried out proof of concept where it was found that the proposal allows to generate rapid prototypes and design embedded systems easily.

Ataide et al. [27] use the MDE approach to transform models to source code for microcontrollers, the proposal comprises a set of tools and domain models that are divided into sub-models, which are inputs to a Petri net for code generation. Moreover, they carry out a proof of concept with two examples. The first corresponds to a single time domain, a single microcontroller, while the second example presents three-time domains, where it is exposed to how to generate code for different microcontrollers.

Previous works show different representations of reuse in Arduino. They have different visions, perspectives, and approaches where reuse is carried out in a disorderly, poorly planned, and sometimes ineffective way. In addition, they do not specifically use the SPLs on the industrial robots with Arduino, this being the greatest contribution that this article intends to make.

### 2.4. Educational Robotic and Arduino

The main Computational Thinking skills that are related to cognitive capability are abstraction and decomposition [28]. Abstraction is an inductive process for simplifying, categorizing, and memorizing key information for processing and storage. Decomposition is a deductive process where a complex situation is broken down into smaller and simpler

pieces, reducing the original complexity. The need to decompose and the abstraction gave place to the modularity [16].

Chalmers in [29] investigated how elementary teachers associate robotics and programming learning in their classrooms, quantifying the perceived impact this has on students' computational thinking skills. The results show that the use of robotics kits enables the development of skills that promote problem-solving and group work with elementary students.

Angeli et al. report in [30] a similar result, where Bee-Bots are used for learning computational thinking. The authors show that children from an early age can manifest decomposition as a problem-solving skill, besides recognizing the importance of integrating robotics into early curricula to develop the learning of computational thinking.

## 3. Materials and Methods

This paper presents IRArduino-SPL and evaluates its usefulness as an educative tool. For the construction, we used the essential methods proposed by the SEI SPL Framework [31], besides Small-SPL guidelines [32]. CoMeS-SPL [33] was used for scoping of the proposal based on the opinion of experts in domains underlying industrial robotics. To evaluate the SPL's usefulness, we used the case study method following the Runeson and Höst guidelines [34]. The case study was performed by three groups of mechatronic engineering students (first year) developing software for operating a robotic arm.

The materials used in the research were source codes for programming industrial robots with Arduino collected on the web, analysis tools for code similarity (Python, Code Compare, and SemanticMerge [35]), S.P.L.O.T [36] and FeatureIDE [37] for modeling the domain and variability of the tool.

The following subsections describe the SPL construction (domain engineering) and its application (product engineering) in the educative context.

### 3.1. Domain Engineering

#### 3.1.1. Arduino Code Analysis

Analyzing an SPL requires scoping and abstracting the domain model and its variability [38]. Thus, we study how industrial robots programs with Arduino are structured, looking for relevant software idioms used in this kind of system. We created a lexicographic analyzer of Arduino code using a Python script. This algorithm searches and retrieves frequently structures used in the source code sample (algorithms of industrial robotic systems with Arduino). We used tokens (keywords) for determining how the software of industrial robots is structured [39], determining the recurrent programming idioms in the domain.

To carry out this study, we used Code Compare and Semantic Merge tools statically analyzing 10 IRS source codes of Arduino. The main result is the organization of Arduino programs by structures, values (variables and constants), and functions. In addition, the study found the most used sentences in the programming of industrial robotic systems, which allowed to establish which are the recurrent programming idioms in Arduino for IRS.

The analysis identified that the Arduino program uses a strategy based on templates, particularly one called *BareMinimum*, which is the minimum source code to compile a program on the platform. This template does not present the necessary utility for the industrial robotic systems domain, nor does it provide reuse options for developers or utilities specific to this domain (kinematics or computer vision). Therefore, templates focused on these devices could facilitate reuse in industrial robotic systems.

The analysis found three blocks with frequent statements in the analyzed codes. The first block contains variable declarations, libraries, and headers; these statements are before *void setup()*. In the second block, there are those related to the initialization of variables and ports, the declaration of library objects; this block of code is usually inside the *void setup()*. The third block relates to the logic adjacent to the IRS operation in Arduino; the code in this section determines the trajectories, motion planning, and operation of sensors and actuators; its position in the source code is inside the *void loop()*.

According to the static analysis, industrial robotic programs have two relevant blocks, the first one has the algorithms for controlling the robot (called, *algorithm block*), and the other one has those for programming hardware devices (called, *hardware block*). The block of algorithms contains the logic for operating the robot, expressed through a pattern of action and reaction: for each stimulus perceived by the sensors, there is an associated action using the actuators. The sentences analyzed in this block lack abstractions, such as kinematics or user interface. The second block of sentences codes the drivers for the correct operation of the physical devices; here, the developers strive to make the sensors and actuators work in accordance and synchronize with the algorithm block.

An element of relevance that was not observed in the analyzed codes is the predetermined functions or preset recipes of the robots, i.e., the functionalities that can commonly fulfill this type of robot for a specific domain (transport of objects, assemblies, and precision work). This type of high-level abstraction could be a contribution to reuse in the area, since it would allow the developer to simply select the functionality and reuse it as many times as necessary. That is why within the abstractions that have been made in the domain, the function called 'activities' has been added, which tries to abstract the common functionalities that industrial robots can use in Arduino so that it can reuse them. The above is not a novel concept in robotic arms used in industry [40], but it is in those in the Arduino domain.

The static analysis also unveiled four fundamental elements within the construction of the software of industrial robotic systems: sensors for detecting environmental changes, movements to perform actions, the joints that have to do with the extension of the system, and the functions of the electro-mechanical device. Thus, its elements were abstracted in the domain's modeling too.

### 3.1.2. Scoping of IRArduino-SPL

To define the scoping of the software product line, CoMeS-SPL was used, which is a collaborative method that guides the definition of this type of reuse strategy. The primary aim of CoMeS-SPL is to strengthen collaboration between different roles and mitigate interdisciplinarity problems with participants with particular interests, using elements such as thinklets and a facilitation process [41]. Three university professors, experts in industrial robot programming, performed coMeS-SPL for determining the SPL Scope. The experts have experience building industrial robots with Arduino and teaching robotics and microcontrollers. The method was executed in two meetings using virtual tools and digital media because of the current circumstances regarding the COVID-19 pandemic [41]; the meetings were synchronous with a duration of one hour and a half. The tools used were Google Meet to organize and conduct the meetings; Lucidchart as a collaborative tool to change, add and delete features to the proposed feature model; and Google Docs for the associated documentation.

1.  *Feature Model:* The IRArduino-SPL feature model was defined using FeatureIDE. Figure 1 shows the IRArduino-SPL feature model where commons (named *Nucleo_Robot*) and variability features (named *Personalizacion_Robot*) have been organized. Common features include joints, sensors, and movements. The *robot_action* feature is mandatory for executing each configuration made and triggering the robot. There is also the *addActivity* feature that endows the robot with a set of sequential logics between conditions and motions, allowing the user to save default logics or add new ones to the robot.
2.  *Feature Model Analysis:* Using S.P.L.O.T and FeatureIDE a syntactic analysis of the IRArduino-SPL feature model was performed, it shows there are 24 features (20 concrete and 4 abstracts) in the model, of which 6 are mandatory, 8 optional, 3 alternatives, and 9 grouped. Likewise, there are 12 composite features and 12 terminal features (leaf nodes). The feature model does not have constraints because specific elements, such as multitasking or computer vision, were left out when defining the scoping of the proposal. After all, according to experts, this type of utility difficult the development

of industrial IRS in Arduino, so it is a feature that must be carefully implemented in future versions of the SPL. It performed a semantic analysis on the feature model, which ruled that it was consistent and valid for both tools. In addition, no dead features were presented. It identified seven core assets as the most important reusable elements of the SPL. In addition, it was established that there are 13,440 valid product configurations. The degree of variability of the proposal is $8.0109 \times 10^{-2}$, which means a lower development cost for each product. Moreover, it found that it must make a minimum of 7 decisions to have a functional derivation. Unlike the previous iteration, this one presented one atomic set, showing that the observed separation between hardware and software does not occur in this version. Finally, a statistical summary of the proposed feature model is presented in the Table 1.

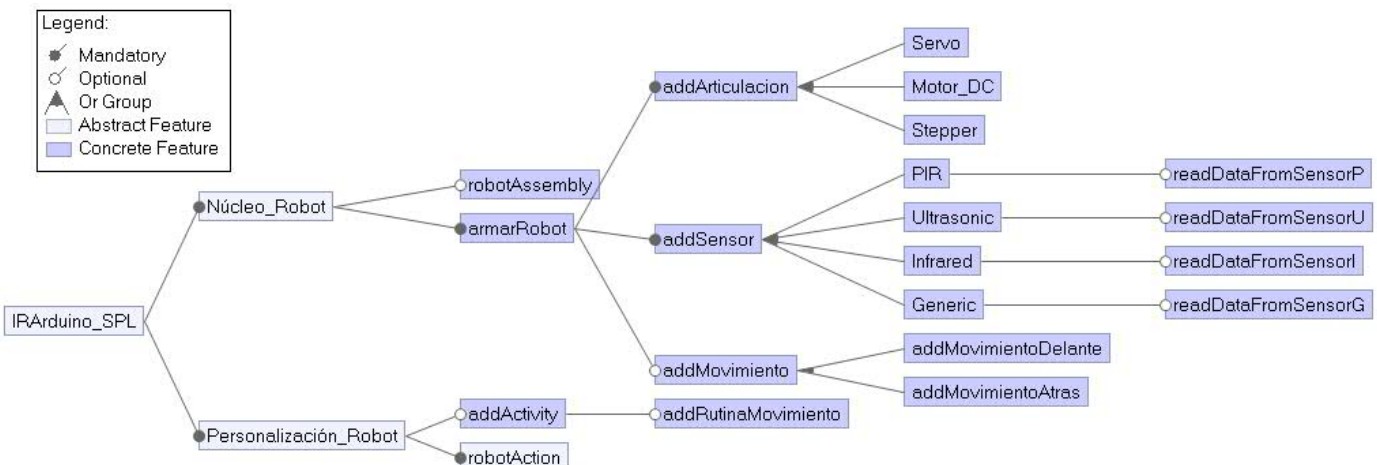

**Figure 1.** Feature model proposed for the SPL developed for industrial robots with Arduino.

**Table 1.** Syntactic and semantic statistics of the feature model for IRArduino-SPL.

| Concept | Number |
|---|---|
| Total characteristics | 24 |
| Concrete characteristics | 20 |
| Abstract characteristics | 4 |
| Composite characteristics | 12 |
| Terminal characteristics | 12 |
| Grouped characteristics | 9 |
| Alternative characteristics | 3 |
| Optional characteristics | 8 |
| Core assets | 7 |
| Dead characteristics | 0 |
| Atomic sets | 1 |
| Valid product configurations | 13.340 |
| Degree of variability (%) | $8.0109 \times 10^{-2}$ |

According to the defined scope, the professors' opinions about the proposal were positive, highlighting the SPL as a novel research topic that may allow the reuse approaches to be useful for learning proposals. They also point out that empirical evidence in the Arduino domain regarding reuse is lacking and that it is a neglected field by the research community. Focusing on the variability of the SPL, the experts were wondering about the many potential products helping the maintainability of the code, so they have expressed several constraints to apply to the feature model. In addition, they suggested a pipeline for customizing components for the device's source code. Thus, students can perceive the concept of mass customization of the software in the reuse strategy. In addition, they stress the importance of focusing on a single type of industrial robot because they consider

the domain must be broad and suggest focusing on three-, four- and five-degrees-of-freedom robotic arms. The microcontroller expert agrees with the use of the object-oriented programming (OOP) paradigm because it is realistic and resembles how humans reason. Each object in the program could simulate the objects (nouns) of the domain (e.g., ultrasonic, servomotor, and gripper, among others) and the methods and actions (verbs, e.g., send a signal, rotate, among others) that the devices can perform. It is a powerful tool for students to learn to program arms, taking advantage of software reuse. Another notable expert's opinion was related to sensors and actuators. He suggested that the peripherals, ports, and communication protocols must be properly defined through code, to organize code associated with the hardware consistently.

### 3.1.3. IRArduino-SPL Architecture

The design uses an object-oriented framework architecture allowing to:

- Abstract common structures and behaviors for implementing with variability.
- Use the polymorphic capabilities to achieve flexibility for composing concrete modules.
- Reuse common structures and behaviors with a new code using the inverted control principle.

The framework created evidence of the reusability in IRArduino-SPL. This framework allows developers of industrial robots to easily develop software through OOP, allowing them to emulate the real entities of the domain being modeled (industrial robots with Arduino). The framework, as Figure 2 shows, provides several high-level abstractions that allow the builder to employ common elements and terms in industrial robot development.

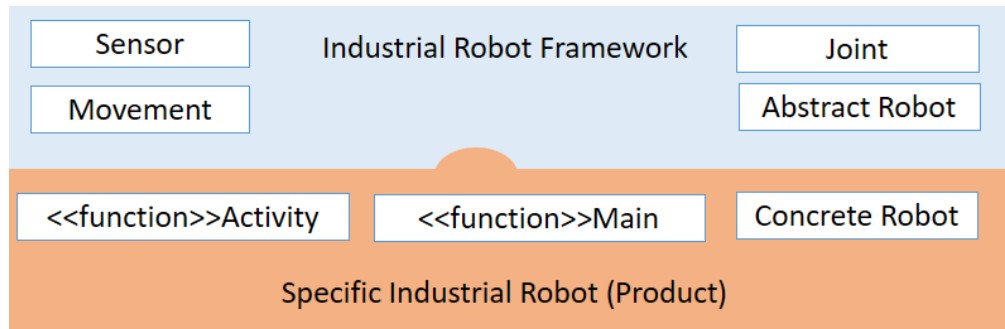

**Figure 2.** IRArduino-SPL framework.

The framework simplifies the development processes. It implements routines and predetermined activities, using a simple architecture for programming industrial robot software with Arduino. The framework's design follows the open-closed principle, allowing to extend its behavior for creating or changing specific programs (improving the maintainability of IRArduino-SPL).

### 3.1.4. Core Asset Development

The high-level abstractions of the framework are built based on the expressed domain by the experts. The main abstraction of the framework was called *RobotIndustrial*, which is an abstract class composed of elements such as joints, sensors, and movements that are stored in a vector for each instantiated object (industrial robots with Arduino), each of these data structures stores the configurations of each robot offering customization capabilities according to the developer's needs. Figure 3 shows these domain assets specified as a class diagram using the Unified Modeling Language, as a result of the domain model abstracted and presented from all domain engineering activities and reflecting the design stage.

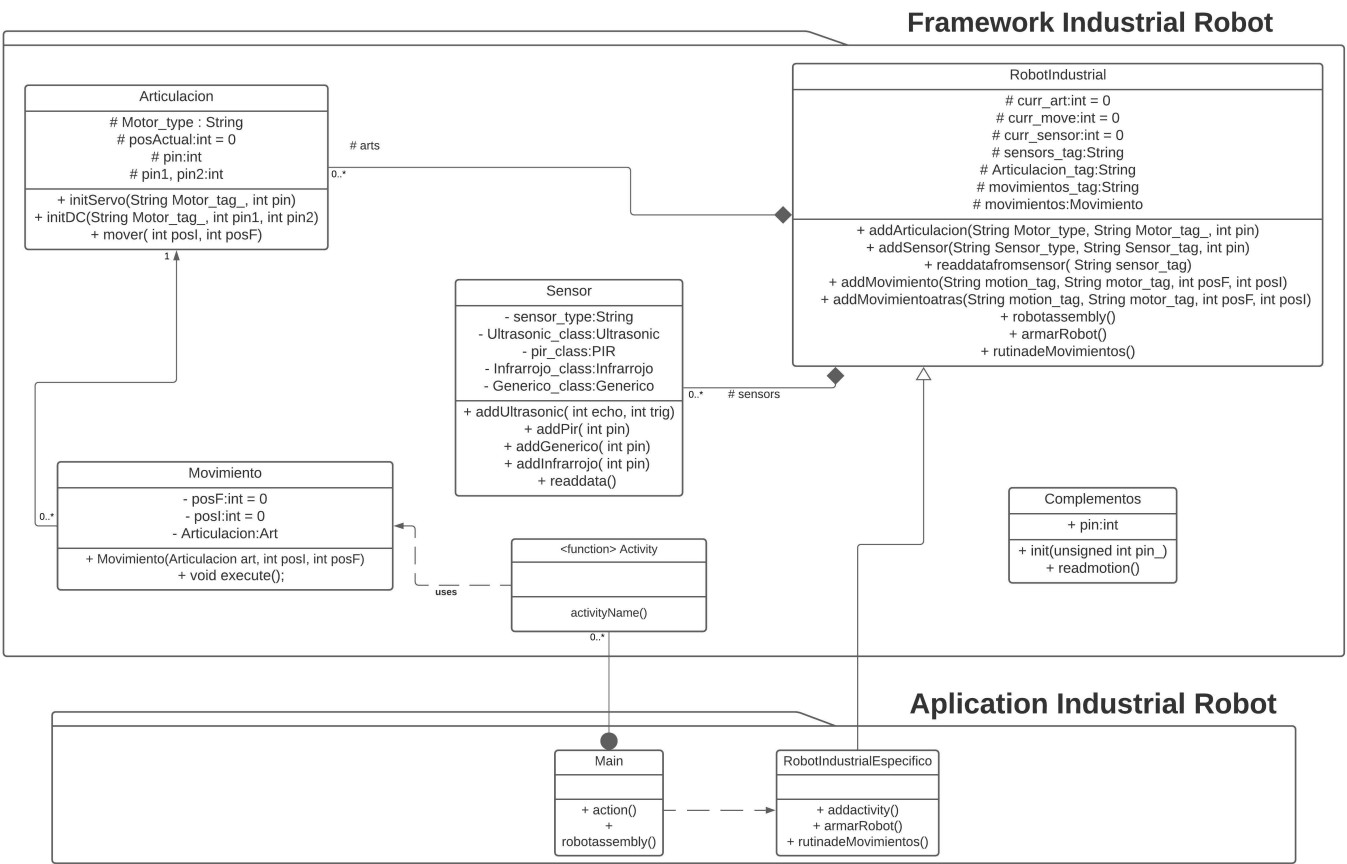

**Figure 3.** A conceptual model of the abstractions made about IRS with Arduino through a class diagram.

The *RobotIndustrial* class has a series of constructors and public methods that allow adding elements such as sensors, actuators, movements, joints, and actions of the robot depending on what the developer requires. Likewise, there are some classes derived from *RobotIndustrial* that are part of the abstractions made on the domain. These are *Complement*, *Movement*, *Articulation*, *Sensor*, and *RobotIndustrialspecifico*. Specifically, the Motion class sends the execution order of the actions performed by the robot actuators; in this case, the servos that allow the displacement of the structure need an initial position and a final position, as well as the joint to which they belong. This class provides access to the *execute()* method that sends the system movement order and allows the library user to define the movement parameters of the actuators and the joint to which they belong. There is also another class defined within the abstractions in the domain designated as *Articulation*. Similar to the physical world and allowing to work with the low-level Arduino hardware, i.e., the framework contains the daily instructions for each actuator without high-level abstractions. Here you can set the pins, names, and current positions of the joints of the devices under construction. Likewise, there is the sensor class which is a generalized abstraction of the sensors (environmental detectors) which is used in industrial robots with Arduino. This uses a global variable that stores the data sent by the detector and executes an action depending on the conditions set by the developer, and it also allows interacting with the daily instructions of the sensors without high-level abstractions (e.g., ultrasonic, PIR, LDR, and photoresistance, among others).

The developer is to extend the framework, instantiating, and assembly objects. The developer must specify the joints, the movements, associations, and the initial and final positions to be executed in the robot. Additionally, the developer can reuse any pre-defined routine of the framework. Figure 4 shows a snippet of the code that supports the creation of software products for IRArduino-SPL.

```
class RobotIndustrial
{
  protected:
    std::vector<Articulacion *> arts;
    std::vector<Sensor *> sensors;
    std::vector<Movimiento *> movimientos;

  public:
    RobotIndustrial() = default;

    virtual void addArticulacion(Articulacion *a)
    {
      arts.push_back(a);
    }

    virtual void addSensor(Sensor *s)
    {
      sensors.push_back(s);
    }

    virtual void addMovimiento(Movimiento *m)
    {
      movimientos.push_back(m);
    }

    virtual void configurarRobot()
    {
      this->armarRobot();
      this->definirRecorrido();
    }

    virtual void action()
    {

      for (int i = 0; i < movimientos.size(); i++)
      {
        movimientos[i]->execute();
      }
    }
    virtual void definirRecorrido() = 0;
    virtual void armarRobot() = 0;
};
```

```
class Robotico
{
  public:

    static void run(void)
    {
      RobotIndustrial *robotico = new RobotIndustrialEspecifico();
      robotico->configurarRobot();
      robotico->action();

      RobotIndustrial *robotico2 = new RobotIndustrial2();
      robotico2->configurarRobot();
      robotico2->action();

      delete robotico;
      delete robotico2;
    }
};
class Movimiento
{
  private:
    Articulacion *art;
    int posI = 0;
    int posF = 0;

  public:
    virtual ~Movimiento()
    {
      delete art;
    }

    Movimiento(Articulacion *art, int posI, int posF)
    {
      this->art = art;
      this->posI = posI;
      this->posF = posF;
    }
    virtual void execute()
    {
      art->mover(posI, posF);
    }
};
```

**Figure 4.** Snippets of source code in the framework developed for industrial robots with Arduino as part of IRArduino-SPL.

The developed framework contains the main core assets of IRArduino-SPL (and it itself is a core asset), showing the software reuse performed in the domain and refined through expert opinion. These core assets allow the reuse of code fragments, high-level abstractions, functionalities, and knowledge that were encapsulated as a library and found thanks to the domain engineering performed. Likewise, these reusable assets can be used in products (industrial robots with Arduino) with different capabilities and features, i.e., from a common reusable core, they got new customizable products.

It is worth noting a limitation that influenced the development of the library for Arduino and that has to do with the computational power offered by the platform; it was found that the computational resources to implement functionalities related to OOP such as polymorphism, inheritance, and encapsulation are limited, which translates into the fact that the possibilities and/or functionalities of the library are restricted, limiting the implementation of high-level abstractions to get a context closer to robotics. The above is not something new in the domain because this resource has also affected other studies [42,43]. Therefore, attempts have also been made to implement traditional solutions to this problem, such as parallel computing [44] or fog computing [45], although without the expected acceptance of diffusion. This poses a problem on the Arduino platform that can be addressed in future research and that represents an important limitation when implementing software engineering on this type of development board.

### 3.2. Application Engineering

The SEI framework mentions that there are three main activities in the development of software product lines: development of the core assets, products development, and SPL management activities [46]. This section details the reuse of the core assets for building a specific industrial robot.

#### 3.2.1. Product Development at IRArduino-SPL

Product development comprises physically manufacturing IRArduino-SPL derivatives from the core assets specified above, based on a production plan, to satisfy user require-

ments. The key inputs in product development are the requirements, product line scope, core assets, and production plan to guide the construction of customizable software from reusable assets [47].

The production plan for IRArduino-SPL comprises specifying the requirements for developing a final product. For this purpose, the software requirements specification must give a detailed description of the system's behavior. For this reason, we have designed a base template as a guide element to structure the development of industrial IRSs with Arduino from the IRArduino-SPL. The template allows developers to specify requirements, such as the general description of the device, the scope of the solution, the person in charge of it, among other elements. Therefore, the developer knows which core assets to reuse and extend.

Figure 5 shows the activities to be performed by developers to produce a specific product from IRArduino-SPL. First, the developer must specify the robot requirements, which must detail characteristics, functionality, and the scope of the solution. Based on this information, the developer structures and organizes the project to be developed. Then, once the requirements are specified, the framework is reused. It gives access to a series of abstractions and high-level functionalities (core assets) and customizes them to build the desired product. Some components can be reused directly; however, the framework allows for programming new hardware (e.g., new types of sensors) and software elements (e.g., creation of new predetermined routes).

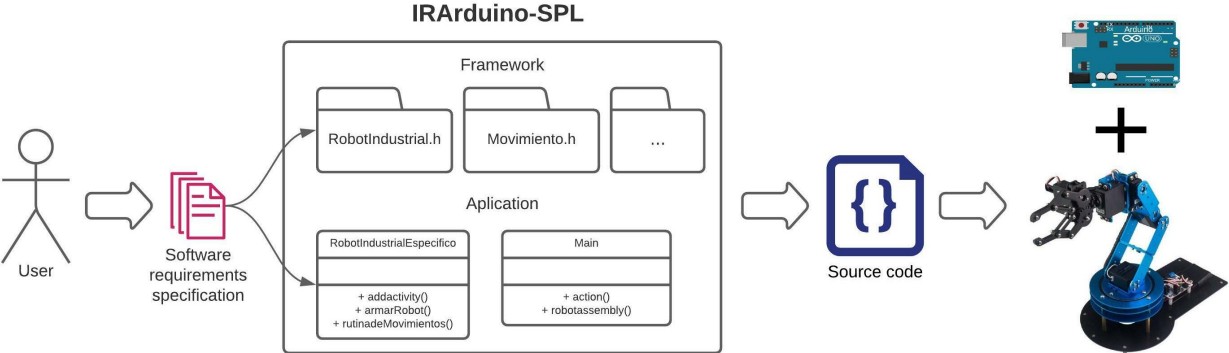

**Figure 5.** Generic IRArduino-SPL use case diagram for production of a product in the SPL.

Subsequently, once the developer has customized the industrial robot, the source code must be charged in the Arduino IDE, which allows the use of its compiler and its communication interface to finally send the code to the industrial robot and perform the relevant tests.

### 3.2.2. IRArduino-SPL Management

Organizing the SPL team development is a complex process that requires additional effort. The entire process has been in charge of the IDIS research group, professors, and students. This research provided an example application for facilitating the learning process for programming robots solutions with Arduino. It allows students to discover the characteristics of the framework and its main functionalities via abstraction and modularity.

The IRArduino-SPL framework follows a gray-box reuse approach. Therefore, the source code can be extended or changed. There are no restrictions on including new hardware elements, especially regarding new types of sensors and actuators. The extensibility mechanism allows for adding many hardware devices and the use of idioms found in domain engineering for both sensors and actuators. To add new sensors in the framework, developers must access the .h file called complements, which is a slot reserved for adding elements required by the programmer, besides being the only area of the library where the developer can interact with the low-level hardware of the platform. For this, you must encapsulate each element to add (sensor or actuator) in a class that defines the properties (pin, configuration, name) and functionalities (sent reading signal) of each sensor in addition to the adjacent logic, if necessary.

Figure 6 shows an example of implementation for a PIR sensor in the proposed SPL, also highlighting the programming idioms found in the domain engineering, which allow simplifying and connecting with the abstractions made in the CAD process. This reuses the logic implemented in the framework, allowing the use of common sentences for all the sensors added and, therefore, the use of a common programming language among all the devices that work under IRArduino-SPL. It enables to use of sentences such as *readdatafromsensor()* independently of the device. The major advantage of this is that it provides an appropriate level of abstraction for the industrial IRS domain with Arduino and takes it away the concerns of low-level hardware and its programming.

```cpp
class PIR {

  public:
    void init(unsigned int pin_) {
      pin = pin_;
      pinMode(pin, INPUT);                    ......> body
    }
    bool readmotion() {
      unsigned int  state = digitalRead(pin);  ....> footer
      if (state == HIGH) {
        return true;
      }
      else {
        return false;                          ......> logic
      }
    }

  private:
    unsigned int pin;                          ......> header
};
```

**Figure 6.** Example of a class that abstracts a PIR sensor for IRArduino-SPL where the main programming idioms encountered in domain engineering are highlighted.

Another activity that takes place within the management of an SPL at the corporate level is the so-called technology foresight, which has to do with making sure that it positioned derived and planned products to take advantage of upcoming technology trends [48]. How could IRArduino-SPL support other technologies? and how could it integrate them into future iterations? A first opportunity is given by the possibility of integrating the Robot Operating System (ROS) as an established technology that could favorably affect the domain of industrial IRS. If IRArduino was implemented with ROS, it could add great technological advantages to the platform (3D simulation, dynamic motion determination, computer vision, support for large grain reuse, and visual odometry systems, among others). This is reflected in [10], where it is expressed that in the general domain of industrial robots, the integration with ROS has been extremely successful, so specific courses on this technology should make students aware of this framework.

Another technological trend that could substantially improve the application domain of IRArduino-SPL is the concept of Smart Factories, which are collaborative manufacturing systems that work in real-time to produce customized goods; this notion encompasses areas of knowledge such as the internet of things, artificial intelligence, and massive data analytics [49]. Smart Factories would broaden the scope of the proposed SPL because it would move from creating tailor-made IRS to implementing a set of collaborative industrial robots working under a common goal. The main advantage of this would be that the learner using IRArduino-SPL would not only understand the concept of software development for robots but would also understand a broader concept such as smart factories with elements of collaborative engineering, big data, and IoT, as well as perceive a leading technology in the domain [50].

To allow replication of the methods used and/or further research, the following link (https://bit.ly/3miPRzv, accessed on 27 December 2021) provides the resources and material used in this research.

## 4. Case Study, Results and Discussion

A case study (Figure 7) based on the guidelines proposed in [34] allows to evaluate with mechatronic engineering students the implementation of IRArduino-SPL. The case is a robotic arm (Figure 8) with 5 degrees of freedom, 2 types of sensors, and 1 robotic gripper, where the system must be derived from the core assets by students following the guideline proposed SPL. It was carried out in a scholarly context in which students from the mechatronics engineering career of the Corporación Universitaria Comfacauca took part. Specifically, the collaboration involves 9 students (divided into 3 random groups) who develop, socialize, implement, and evaluate the solution. The selected students know areas such as microcontrollers, software programming, and the development of robotic systems in hardware and software, so they know the areas needed to build an IRS with Arduino. The activities that were followed during the case study were: socialization and contextualization of the case study, presentation and introduction to IRArduino-SPL through examples, application of the strategy proposed by the students, and finally the evaluation of the application and a satisfaction survey to measure the usefulness of the tool.

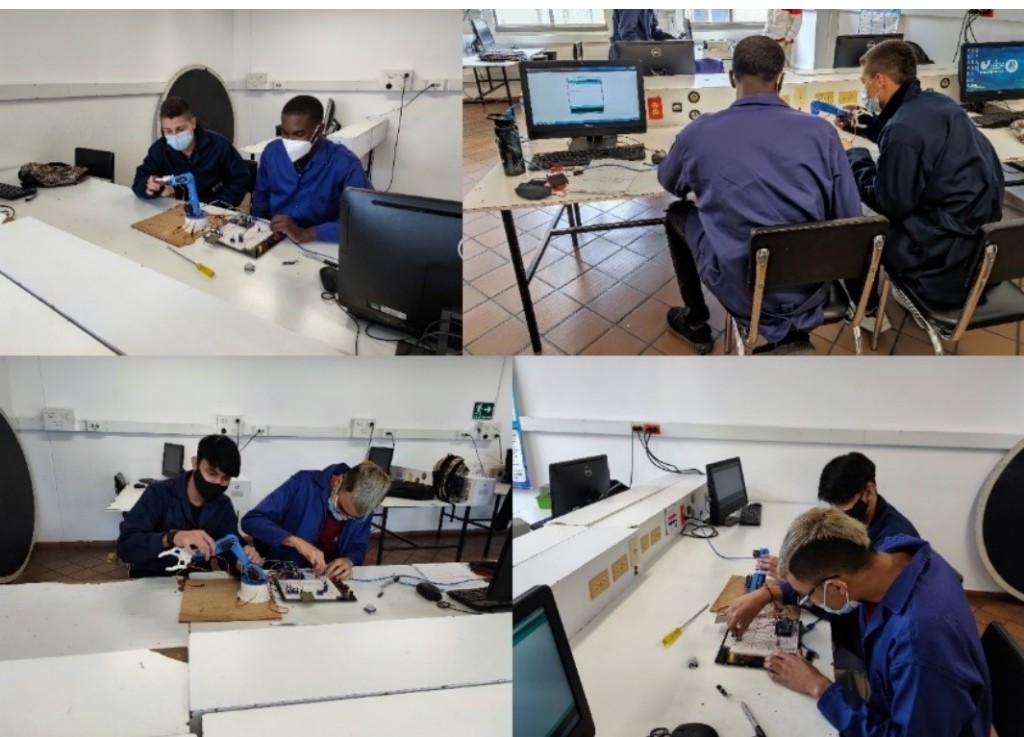

**Figure 7.** Photographic evidence of the random groups that developed the case study using IRArduino-SPL for the software development of an industrial robotics system with Arduino.

The students are to program a robot arm for transporting an object from one place to another depending on changes in the environment (perceived by two sensors), using the IRArduino-SPL framework. To avoid hardware influencing the results of the case study, the same hardware was used to test each product derived from IRArduino-SPL by each group, ensuring that hardware did not bias the results.

To determine the usefulness of IRArduino-SPL in the case study, it quantified the software reuse index in each of the taking part groups. This reusability index is composed of a set of three metrics that allow determining the actual usefulness of the tool concerning the software required by the industrial robots developed. The reusability index is composed of three metrics: the percentage of software reuse (R) proposed in [51], the Lines of Feature Code (*LoF*), and the Number of Features (*NoF*) proposed in [52]. In addition, to evaluate the usability perceived by the users who employed IRArduino-SPL, a 16-question questionnaire was applied, which is based on the standardized questionnaires of Sauro et al. in [53].

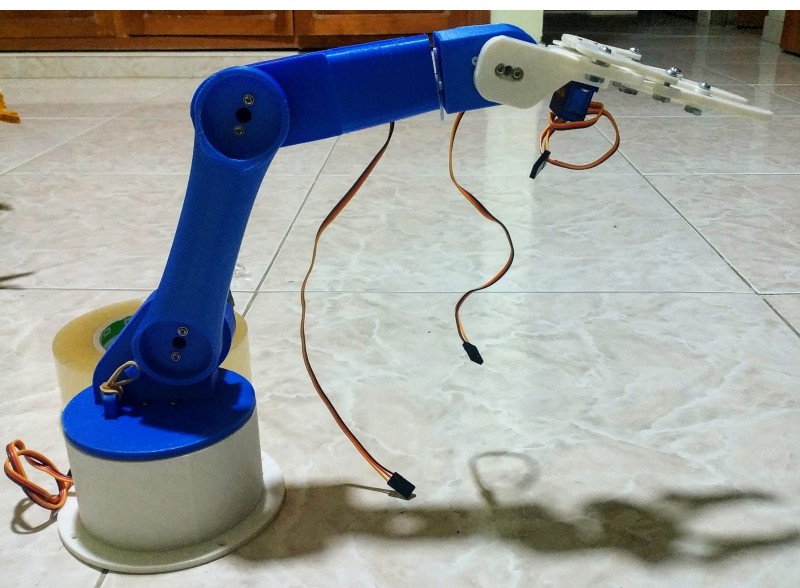

**Figure 8.** Robotic arm that was used for the execution of the case study.

### 4.1. Case Study Execution and Data Collection

This section shows the execution and the record of the data taken from the case study, where the students developed the software for the operation of an industrial robotic system based on the guidelines provided and IRArduino-SPL. Table 2 shows the information collected regarding the time required by each group to develop the software for this type of electromechanical device. It measured the time taken by the activities in hours.

**Table 2.** Time records for the execution of the case study with IRArduino-SPL.

| X | Software Development Time | Total Time for the Development of the Case Study |
|---|---|---|
| Group 1 | 2.0 | 2.3 |
| Group 2 | 2.3 | 2.5 |
| Group 3 | 2.1 | 2.3 |

The time spent by the participants in the execution of the case study indicates that for two of the three groups, it was not enough. It could be a sign that the tool has a steep learning curve and that the students need more time to understand the abstractions made about the domain and the application of the tool. Students had doubts in understanding the logic proposed by the developers for IRArduino-SPL because they needed clarifications on the proposed abstractions, the methods to introduce them, as well as, a conditioning time to reorient the way they usually develop (procedural) for hardware towards an object-oriented programming approach. The above may be reasons the students took longer than expected in the case study which could also be improved in future iterations of SPL training.

### 4.2. Metric 1: Percentage of Software Reuse

The percentage of software reuse (Equation (1)) estimates the number of lines of code (without comments or blank lines) for a product P generated from IRArduino-SPL in percentage terms [51].

$$R = (LR/TL) * 100 \tag{1}$$

where *LR* is the number of reused Source Lines of Code (SLOC), *TL* is the total lines of code in the system, and *R* is the software reusability of the proposal in terms of a percentage of the total lines of code. At a general level, results show that IRArduino-SPL is useful because each of the three groups derived a product from the framework. Specifically, in the first group, the percentage of reuse was 43%; for the second group it was 36%, and

for the third group, it was 38%. The framework allows for speeding up the development of software applications for industrial robotic systems with Arduino. Table 3 shows the percentages of software reuse in the case study.

**Table 3.** Records of the percentage of software reuse in the execution of the IRArduino-SPL case study.

| X | Percentage of Software Reuse | Total SLOC |
|---|---|---|
| Group 1 | 43% | 423 |
| Group 2 | 36% | 385 |
| Group 3 | 38% | 391 |

Focusing the analysis on the total SLOC of each of the groups, it can be observed at a general level that the variation is 38 lines of code between groups 1 and 2. This may be because the first ones program an extra movement (not required) to balance the robot and that the servomotors (actuators) do not lose torque. Groups 2 and 3 have a similar percentage of reuse presenting a difference of 2%, as in the total number of lines of code. This small contrast is manifested because group 2 in the selection of sensors opted for a gas sensor that is not in the repository of IRArduino-SPL, so they had to create the code for this sensor using the logic of the proposed tool, unlike group 3 that selected two sensors (PIR sensor and ultrasonic sensor) that were already in the repository. It shows the capability of the framework for covering a great variety of reuse scenarios, allowing teachers and students to explore many possible implementations.

### 4.3. Metric 2: Lines of Feature Code

The LoF metric allows us to determine whether a small fraction of the source code is variable or not. A high value indicates that there are many lines of code dedicated to the realization of the feature. The calculated value reveals the amount of variable code and thus the complexity of maintainability [52]. The following equation allows determining the mentioned metric, where *LRF* is the number of lines of feature code that are linked to a feature, and *TL* is the number of total lines of the developed software.

$$LoF = (LRF/TL) * 100 \tag{2}$$

They are a few source code statements related to the feature calls. This indicates that the most implemented features in the case study are those with a lower level of abstraction, for example, features like Servo, Ultrasonic, *addMotionForward*, or *addMotionBack*, which could mean that to achieve higher levels of reusability in the code, these specific statements should be abstracted to a higher level, and linked to a hardware element. Table 4 shows the values found for this metric.

**Table 4.** Lines of code of features quantified in the execution of the case study with IRArduino-SPL.

| X | LoF | SLOC Linked Features |
|---|---|---|
| Group 1 | 16% | 70 |
| Group 2 | 15% | 58 |
| Group 3 | 17% | 60 |

Although indeed, the value got in LoF could not be interpreted as a calculated result, some authors relate this metric with a high value suggesting a high complexity in maintenance tasks, because each feature is a block of code that implements one or more functionalities and its modification can lead to a series of consequences in the developed system. Specifically, if the results of the case study are compared with other studies, the values of the LoF metric are a little higher than normal, with IRArduino-SPL the results show between 15 and 17%, while the consulted researches present values of around 13%, without forgetting that these are product lines implemented only in software-related do-

mains. Therefore, in future iterations, work should be carried out to reduce the value of the metric and thus improve the maintainability of IRArduino-SPL.

### 4.4. Metric 3: Number of Features

The NoF metric is directly related to the number of features used in implementing a software product line. When the value is closer to N, the more features a program has and its maintenance can be more complex [52].

The IRArduino-SPL implementations performed by the groups were diverse because they gave them the possibility to include in their robot different hardware elements supported by the tool or to create some sensors or actuators. It also reflected this in the number of features employed by the students. Therefore, in the NoF metric, some variety in the number of features employed by each group can be observed. The diversity of features used by the groups may also be because of the logic implemented in IRArduino-SPL, where features such as *robotAssembly* or *armarRobot* are necessary for the code to work correctly.

Figure 9 shows sets out the features used by each group in conducting the IRArduino-SPL case study. The images show the features used with a red box and the number of times they were used.

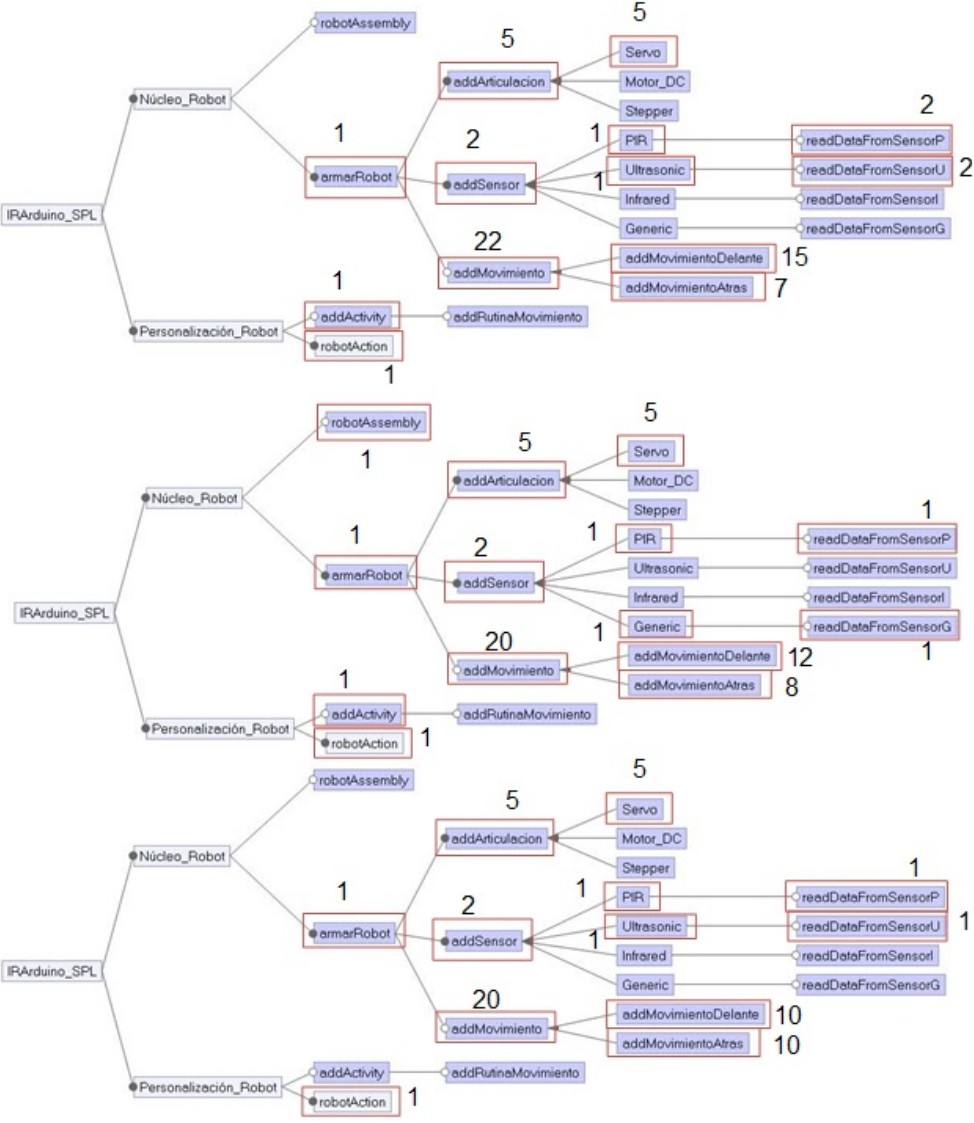

**Figure 9.** Features used by each of the groups in the execution of the case study for IRArduino-SPL.

Specifically, group 1 used 65 feature instances that are a representation of the generated product (industrial SR with Arduino) from IRArduino-SPL for their specific case. This group used a PIR sensor, an ultrasonic sensor, and 5 servo motors. Regarding the sensors, it can be shown that 4 features related to the data reading functionalities (*readDataFromSensorP* and *readDataFromSensorU*) are also used, which could in future iterations not be a separate feature, but be included within the sensor feature, so as not to complicate the maintainability of the SPL. Moreover, it can be seen that the *addActivity* feature was used, which is linked to the recurring activities that the robots do (movements) after detecting a change in the environment (sensor). Here, the students used this feature so that once the PIR sensor detects something; the robot makes a series of pre-programmed movements that have to do with grabbing an object and leaving the robot in a central position of its axes, allowing to save lines of source code so that the robot returns to a central position after grabbing the object.

For group 2 it can be observed that the number of feature instances (60) are like those of group 1, with only 5 features difference, this is because the sensors of this group census twice to ensure that the detection is correct which does not occur for group 2, this is related more to the underlying hardware and its accuracy and not to the robot code. Moreover, students in group 2 used a similar logic to group 1, using the *addActivity* feature they add predetermined activities when the robot detects a change in the environment. Finally, it should be noted that it used the generic feature in sensors because the students wanted to use a gas sensor that was not among the options provided in IRArduino-SPL.

Group 3 used 58 feature instances, which compared to the other groups, is the smallest number, which is striking since they also got the lowest percentage of reuse. Therefore, it could be shown that for IRArduino-SPL, the more features used, the higher the reusability, but product maintainability may be more complicated.

## 4.5. Usability at IRArduino-SPL

Students find it easy to use IRArduino-SPL, because it uses common terms of the domain such as sensors, actuators, joints, among others, providing an advantage over the normal Arduino environment. However, some difficulties were observed in the technical knowledge of OOP, where some students do not understand facilities such as inheritance or polymorphism that are exploited within the proposed reuse strategy. Moreover, it was detected with the case study that IRArduino-SPL has a steep learning curve because the students had some problems with the understanding of some software engineering concepts, such as abstraction, modularity, product lines, or code generators. After all, these types of tools are little used or spread in students of hardware-related careers, such as electronic engineering or mechatronics engineering, so students should be introduced to this type of facilities for software development, such as ROS or the same proposal of this research.

Among the different perspectives expressed by the students, it was found that IRArduino-SPL provides some advantages that have to do with the use of object-oriented programming since it allowed choosing among different industrial robots with Arduino that have some similar characteristics. Using high-level abstractions has allowed us to break down some of the most common functions used within the Arduino ecosystem in the SRI, such as transporting objects or reacting to a change in the environment, allowing these to be reused in future constructions of industrial robots. Another important functionality is the use of a sensor and actuator repository that provides developers with functions specific to these devices and acts as a hardware abstraction layer in the domain.

The current limitations of the Arduino development platform do not allow adopting specific elements of IRS such as graphical interfaces or motion kinematics, so future research derived from this work may be related to porting IRArduino-SPL to more specialized software such as ROS allowing to count on the facilities offered by Arduino (ease of development, compatibility, free hardware), but focusing on the specific domain of robots using these microcontrollers so that special emphasis is placed on elements such as virtualization, motion dynamics, signal conditioning, and software reuse, among other specific aspects of these electromechanical systems.

## 5. Conclusions and Further Work

IRArduino-SPL is a software product line for educational purposes. IRArduino-SPL development includes domain and application engineering activities for building industrial robots with Arduino. The reusable assets, such as domain model, feature model, scope, and production strategy, are the basis for the SPL. In addition, a case study was developed applying the SPL with mechatronic engineering students, using Arduino with a robotic arm with 5 degrees of freedom, 2 types of sensors, and 1 robotic gripper for the Arduino platform. As a major result, it was found that IRArduino-SPL is a coherent, useful, and implementable reuse strategy in academic environments. Thus, the SPL abstraction and modularity enable it to be built by reusing generic elements. The major contribution of this research is the empirical evidence using specific reuse approaches, such as SPLs, in a little-explored domain from the point of view of software reuse and its opportunity in an educational context. Implementing IRArduino-SPL contributes to the Arduino domain, with knowledge and experience where evidence is limited and unclear. Additionally, the robotic programming learner can establish the basis for the future development of robots in industrial settings.

The analysis, focusing on the metrics studied in the case study, shows that a software reuse mechanism based on engineering product lines improved the software development reuse by approximately 40%. It represents less programming effort and raises the abstraction level to understand and achieve decomposition in the SPL. Therefore, the efforts saved by students could be used for learning other matters in both robotic and computer science.

As future work, integrating technologies such as ROS or Gazebo could enable some functionalities that the Arduino domain limits because of its computational capacities, such as artificial vision or the determination of robot mobility dynamics. Moreover, the evolutionary processes of SPLs can take years, so new reuse approaches, such as model-driven engineering or component-based development, can be added for achieving new interactions.

**Author Contributions:** Conceptualization, A.F.S.P. and J.A.H.A.; methodology, A.F.S.P., P.H.R. and J.A.H.A.; software, A.F.S.P. and J.A.H.A.; validation, A.F.S.P. and P.H.R.; formal analysis, A.F.S.P., P.H.R. and J.A.H.A.; writing—review and editing, A.F.S.P., P.H.R. and J.A.H.A. All authors have read and agreed to the published version of the manuscript.

**Funding:** This research received no external funding, and the APC was funded by Corporación Universitaria Comfacauca and Universidad del Cauca.

**Institutional Review Board Statement:** Not applicable.

**Informed Consent Statement:** Informed consent was obtained from all subjects involved in the study.

**Data Availability Statement:** A video of the proof of concept and materials used in this research can be accessed in the following link (https://bit.ly/3miPRzv, accessed on 27 December 2021).

**Acknowledgments:** Thanks to Corporación Universitaria Comfacauca for providing its facilities and to Jorge Prado and Ginna Andrea Ramírez for their help in conducting the research.

**Conflicts of Interest:** The authors declare no conflict of interest.

## Abbreviations

The following abbreviations are used in this manuscript:

| | |
|---|---|
| SPL | Software Product Line |
| IRS | Industrial Robotic Systems |
| MDE | Model-Driven Engineering |
| CBSE | Component-Based Software Engineering |
| SOA | Service Oriented Architecture |
| OOP | Object-oriented programming |
| SLOC | Source Lines of Code |
| ROS | Robot Operating System |

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
