# Peer review of "A Software Products Line as Educational Tool to Learn Industrial Robots Programming with Arduino"

_electronics, doi:10.3390/electronics11050769_

Round 1

Reviewer 1 Report

Dear Authors,

I have read your well-written manuscript untiled "A Software Products Line for Industrial Robots with Arduino in an Educational Context: IRArduino-SPL."

An important and well-defined research problem, unique research area, the impact of the research performed on the existing state of arts are good.

The quality of planning and conducting the research, criteria for selecting data, the paper's theoretical framework, research methods, and their correct selection and application are good.

A clear layout, cohesion between the main purpose, title and content of the paper, results coming from methods used in the article are good.

The clear and precise language, correct terminology, the accuracy of references, articles from reputable journals used in the paper are good.

I recommend your manuscript for publication in the International Journal of Environmental Research and Public Health.

Thank you and good luck!

The reviewer

Author Response

Popayán, December 22, 2021

Dear,

Editor in Chief

Journal Electronics

I kindly submit the corrected version of the paper entitled "A Software Products Line for Industrial Robots with Arduino in an Educational Context: IRArduino-SPL", according to the suggestions and comments of the reviewers. The respective corrections or clarifications were made. The attached table shows each of the changes made to the evaluators' comments. I will be pending any additional requirements you may consider pertinent.

Sincerely,

___________________________

MSc. Andrés Felipe Solis Pino

Autor de correspondencia

Corporación Universitaria Comfacauca

Evaluator's comment 1

SUGGESTION ACCEPTED OR REJECTED

ANSWER

The reviewer states that the article is suitable for publication, so he has no suggestions about the document.

Evaluator's comment 2

Answer

They should rework the title of their manuscript in order to better reflect the objectives of their work.

Accepted

Following the reviewer's suggestion, the title of the article was modified, focusing it on education, as is the objective of the issue for publication, without losing sight of software product line engineering as a fundamental part of the proposal.

The same with the abstract, which should have more clarity and the less utilization of the word “reuse” in it.

Accepted

Following the reviewer's suggestion, the summary of the article was modified, focusing it on education, as is the objective of the issue for publication, without losing sight of software product line engineering as a fundamental part of the proposal, the methods used, and the main conclusions.

The ‘Introduction’ section is full of good arguments but lacks the necessary punctuality, mainly due to language style reasons.

Accepted

The "Introduction" section was modified to be more specific in the justification of the research.

The third section describes the materials and methods used in the research, including a study of the results found in SPL basic construction.” The contribution of the specific work against works previously done should be better clarified.

Accepted

A specific paragraph was added in the introduction and in the conclusions where the main contributions of the work are mentioned. Example “The major contribution of this research is to generate empirical evidence of a specific reuse proposal, such as SPLs, in a domain little explored from the point of view of software reuse and its opportunity in an educational context”

To better highlight the importance of using arduino-inspired tools for the delivery of working robotic solutions, before any analytic SPL solution description, the ‘Related Work’ section should be enriched with specific characteristic arduino-based robotic design examples (e.g., for robotic arms, vehicles, etc.). The role of visual programming add-ons for the Arduino IDE in the standardization of the robotic programming should be also mentioned. Towards this direction, the following references can be useful:

Accepted

Some of the references suggested by the reviewer were added, in addition to others that have relevance within the field of robotics with Arduino. Likewise, some researches that relate computational thinking and educational robotics were added.

Some paragraphs (e.g., in Section 3) are too extended and should be split, in order to become more analytically comprehensive to the non-expert reader. On the other hand, they should also avoid to use one-sentence paragraphs.

Accepted

To improve the wording of the paragraphs, they were divided in such a way as to make them easier for readers to read.

They should provide detailed information and references for every instrument (i.e., mainly assessment tool) they used in their research.

Accepted

Finally, to enable replication of the methods used and/or further research, a link is provided where resources and material related to this work (surveys, S.P.L.O.T data analysis, photographic evidence, etc.) can be found.

They should avoid to use first and second person in their expressions.

Accepted

Corrected use of first persons in the text

It is important the authors should provide details concerning their contribution/or plans for an easy-to-use graphical user interface for the IRArduino-SPL tool.

Accepted

Future work includes the integration of new technologies such as ROS or Acceleo to improve IRArduino-SPL implementations, with special emphasis on the graphical interface and new technologies.

Considerable English language style and grammar issues should be addressed throughout the manuscript (e.g., to select the correct preposition after main verb type, or to use the proper synonyms).

Accepted

The general wording of the document has been improved, with special emphasis on some software engineering terms, and native English teachers were consulted to revise the language.

Evaluator's comment 3

Answer

Be more precise and mention specific industries where this work is usefull

Accepted

A specific paragraph was added in the introduction and in the conclusions where the main contributions of the work are mentioned. Example “The major contribution of this research is to generate empirical evidence of a specific reuse proposal, such as SPLs, in a domain little explored from the point of view of software reuse and its opportunity in an educational context”

language needs improvement a some places.

Accepted

The general wording of the document has been improved, with special emphasis on some software engineering terms, and native English teachers were consulted to revise the language.

The list of references is long, and some references are not cited. So, citem or delete them.

Accepted

The list of references has been rectified and all are cited in the text. It should also be mentioned that at the request of another reviewer, the number of references in the related works had to be increased.

Reviewer 2 Report

In the under review papers, the authors are providing details referring to a tool for increasing the software modules reuse degree (and thus the efficiency) in industrial robotic systems programming, starting from student-level implementations.

They work is very interesting and deep, it also provides valuable results, but considerable improvements/reassessment is required to make it mature enough for publication according to the Electronics Journal standards:

  • They should rework the title of their manuscript in order to better reflect the objectives of their work.
  • The same with the abstract, which should have more clarity and the less utilization of the word “reuse” in it.
  • The ‘Introduction’ section is full of good arguments but lacks the necessary punctuality, mainly due to language style reasons.
  • The third section describes the materials and methods used in the research, including a study of the results found in SPL basic construction.” The contribution of the specific work against works previously done should be better clarified.
  • To better highlight the importance of using arduino-inspired tools for the delivery of working robotic solutions, before any analytic SPL solution description, the ‘Related Work’ section should be enriched with specific characteristic arduino-based robotic design examples (e.g., for robotic arms, vehicles, etc. ). The role of visual programming add-ons for the Arduino IDE in the standardization of the robotic programming should be also mentioned. Towards this direction, the following references can be useful:
    • Concha Sánchez, A.; Figueroa-Rodríguez, J.F.; Fuentes-Covarrubias, A.G.; Fuentes-Covarrubias, R.; Gadi, S.K. Recycling and Updating an Educational Robot Manipulator with Open-Hardware-Architecture. Sensors 2020, 20, 1694. https://doi.org/10.3390/s20061694
    • Yacoby, D.; Yehezkel, L.; Inbar, O.; Zarrouk, D. Design and Modeling of a Parent Big STAR Robot Platform That Carries a Child RSTAR. Appl. Sci. 2020, 10, 8767. https://doi.org/10.3390/app10248767
    • Loukatos, D.; Petrongonas, E.; Manes, K.; Kyrtopoulos, I.-V.; Dimou, V.; Arvanitis, K.G. A Synergy of Innovative Technologies towards Implementing an Autonomous DIY Electric Vehicle for Harvester-Assisting Purposes. Machines 2021, 9, 82. https://doi.org/10.3390/machines9040082
    • Loukatos, D.; Templalexis, C.; Lentzou, D.; Xanthopoulos, G.; Arvanitis, K.G. Enhancing a flexible robotic spraying platform for distant plant inspection via high-quality thermal imagery data, Computers and Electronics in Agriculture, Vol. 190, 2021, ISSN 0168-1699. https://doi.org/10.1016/j.compag.2021.106462
    • Kulshreshtha, M.; Chandra, S.S.; Randhawa, P.; Tsaramirsis, G.; Khadidos, A.; Khadidos, A.O. OATCR: Outdoor Autonomous Trash-Collecting Robot Design Using YOLOv4-Tiny. Electronics 2021, 10, 2292. https://doi.org/10.3390/electronics10182292
    • Tsalmpouris, G.; Tsinarakis, G.; Gertsakis, N.; Chatzichristofis, S.A.; Doitsidis, L. HYDRA: Introducing a Low-Cost Framework for STEM Education Using Open Tools. Electronics 2021, 10, 3056. https://doi.org/10.3390/electronics10243056
  • They should provide detailed information and references for every instrument (i.e., mainly assessment tool) they used in their research.
  • Some paragraphs (e.g., in Section 3) are too extended and should be split, in order to become more analytically comprehensive to the non-expert reader. On the other hand, they should also avoid to use one-sentence paragraphs.
  • They should avoid to use first and second person in their expressions.
  • It is important the authors should provide details concerning their contribution/or plans for an easy-to-use graphical user interface for the IRArduino-SPL tool.
  • They should expand (into the future) the evaluation of the proposed tool using more students and/or professionals. The duration for completing the described robotic activity, when students did not use the IRArduino-SPL tool should also be reported.   
  • Considerable English language style and grammar issues should be addressed throughout the manuscript (e.g., to select the correct preposition after main verb type, or to use the proper synonyms).

Author Response

(The authors gave the same response as above.)

Reviewer 3 Report

The paper develops a technique for reusing robots.

Strenghts:(a) The efficiency and feasibility of SPL  is shown.

The idea of software product lines is used.

This improves earlier Arduino type results.

(b)This work is usefull in education.

(c)Future possible  techniques and technologies are suggested for future research.

(d)The Arduino domain of aplications is expanded.

Weaknesses:(a) The list of references is long, and some references are not cited. So, citem or delete them.

(b)Be more precise and mention specific industries where this work is usefull

(say in the introduction).

(c)The language needs improvement a some places.

Author Response

(The authors gave the same response as above.)

Round 2

Reviewer 2 Report

Dear authors, 

You have revised your work satisfactorily. I have no futher comments on it. 

Reviewer 3 Report

Changes made.